# DiZCo: Planning Zero-Shot Coordination in World Models

## Abstract

Developing intelligent agents capable of seamlessly cooperating and coordinating with other agents in shared environments, including humans, has become a critical research challenge in the field of AI. This requires agents to understand environment dynamics and anticipate other agents responses' to each action. Current research approaches Human-AI coordination through model-free policies for optimality and population-based training for robustness. However, these approaches are brittle and can fail when collaborating with people due to the diverse and unpredictable nature of human behavior, which cannot be comprehensively captured by the training distribution. Striving for a solution that balances robustness and optimality, we introduce **DiZCo**[1], the first framework that leverages generative models to enable real-time, search-based planning in a complex human-AI cooperative task. We first train a generative model to predict future world trajectories conditioned on current state, ego actions, and partner identity, serving as our world model. Then we train a generative action proposer that proposes plausible ego action candidates based on the world state. At test time, we identify the optimal future trajectory by searching through outcomes of all proposed action candidates passed into our world model. Offline evaluations indicate that the **DiZCo** framework outperforms state-of-the-art model-free policies in joint reward. To validate that this method can be feasible for real-time human interaction, we engineer a system that enables model-based planning and search to operate at speeds fast enough to cooperative live with humans. A preliminary user study resulted in positive feedback, collectively underscoring its practical effectiveness for real-time human-AI collaboration.

## 1 Introduction

Human collaboration is a real-time negotiation, in which partners continuously infer intentions, resolve conflicting objectives, and adapt to differences in skill, timing, and behavior. Success depends on interpreting implicit signals while pursuing an explicit, shared goal. Therefore, building predictive models that can support effective collaboration is a major challenge for AI systems, as they must account for the wide variability in human skills, goals, and behavior patterns across individuals and contexts. These AI systems must be able to anticipate how the world and their partners evolve in response to each action, both immediately and over extended horizons, to effectively collaborate with humans. Unlocking this would enable a future where intelligent robots would become an integral part of our day-to-day lives, assisting and adapting alongside us.

Current state-of-the-art approaches for human-AI cooperation are predominantly comprised of model-free reinforcement learning (RL) approaches, augmented with population-based training (PBT) (Liang et al., 2024; Zhao et al., 2023). While these have proved able to generalize to in-distribution partners, they are fundamentally limited. These methods learn a reactive policy that implicitly memorizes responses to partner behaviors seen within the training population, struggling to coordinate with partners exhibiting novel strategies or behaviors that lie outside of the distribution. This lack of explicit reasoning and planning makes them brittle when faced with online adaptation a truly unseen partner. Given the level of diversity of human behaviors, brittle methods are funda-

---

[1] Project website: https://sites.google.com/view/dizco-planning/

mentally ill-suited to the problem, and fail to scale and generalize well. Our work addresses this gap by moving towards a model-based approach allowing for test-time inference and planning.

We introduce **DiZCo**, a **Di**ffusion-based framework for **Z**ero-shot **Co**ordination that achieves real-time adaptation through online planning. At its core, diffusion models are capable addressing two key challenges in multi-agent settings: **generalization to an arbitrary number of agents** and **adaptation to novel partners**. Our approach uses two complementary diffusion models: 1) an **action proposal model**, conditioned on historical context and reward-to-go, generates candidate ego-action sequences, and 2) a **partner-conditioned world model** simulates the future outcomes for each candidate sequence. We search over these simulated rollouts, selecting the plan that maximizes the joint reward. For real time evaluation, **DIZCO** searches over the simulated rollouts to select a high reward plan given the partner 1. Employing a diffusion model as the simulator, we effectively disentangle the problem of modeling complex partner and environment dynamics from the separate problem of planning. This separation of search, plan generation and evaluation, allows our agent to reason about partner behavior, resulting in more robust and adaptive strategies than common model-free approaches.

The effectiveness of allocating additional compute at inference time via search is recognized in recent work on large language models (LLMs), where test-time reasoning substantially improves performance (Snell et al., 2024; Liu et al., 2025; Beeching et al.). Analogously, our framework leverages test-time compute using our partner-conditioned world model by allowing the agent to explore potential futures under a given partner behavior, evaluates their expected outcomes, and selects strategies that are robust to variations in partner behavior. Importantly, because DIZCO uses search as its optimization procedure during test time, it does not need to memorize a brittle optimal strategy for all possible partner policies, as is often seen in model-free approaches. Each partner is modeled and searched over individually, respecting the unique differences between partners and enabling true personalized coordination.

Model based approaches have often been overlooked in the human-AI coordination space, due to its computational challenges and potential impracticability for real-time live interaction. However, **DIZCO**, the first framework of its kind, leverages diffusion generative models to enable **real-time, search based planning in a complex human-AI cooperative task**, outperforming current state-of-the-art methods in offline evaluation.

In sum, our contribution includes:

- We introduce **DIZCO**, the first framework leveraging diffusion generative models to enable **real-time, search based planning in a complex human-AI cooperative task**, overcoming long-standing concerns about computational tractability.
- We design a **two-model architecture**: (1) an *action proposer* conditioned on history and reward-to-go, and (2) a *partner-conditioned world model* that simulates futures given candidate actions.
- We show how **test-time search** over simulated rollouts enables **personalized partner modeling**, avoiding the need to memorize brittle strategies across diverse partners, yields **robust and adaptive coordination**.
- We empirically show that DIZCO **outperforms state-of-the-art model-free baselines** in offline evaluation on complex cooperative tasks including **Overcooked** and **Autonomous Driving**.

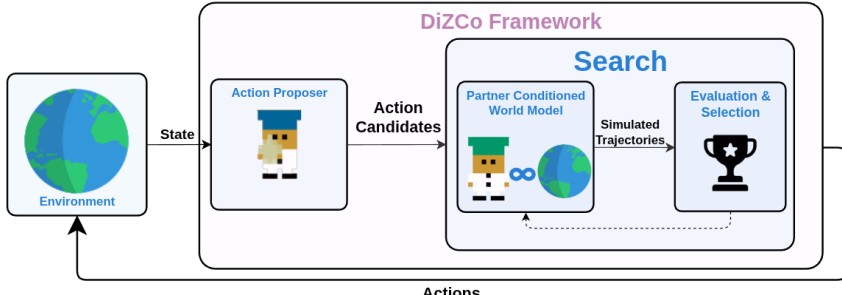

Figure 1: The DIZCO online planning loop: The Action Proposer generates candidate actions, which are passed to a Search module. The Search module then uses the World Model to simulate and score future trajectories, selecting the optimal plan for execution.

## 2 RELATED WORKS

### 2.1 COOPERATION WITH GENERATIVE MODELS

**Cooperation with Generative Models** Developing cooperative agents that generalize effectively to human partners remains a long-standing problem in AI, known as ad-hoc team-play (Stone et al., 2010), or zero-shot coordination (Hu et al., 2020). A central challenge in human-AI cooperation is adapting to the diverse and unpredictable nature of human behavior. Grover et al. (Grover et al., 2018) pioneered the work of generative representation of partner policy through a latent vector. Since then, generative modeling has become a nascent tool to generate a valid and diverse population of training partners for effective human-AI cooperation in various domains (Liang et al., 2024; Chaudhary et al., 2025; Wang et al., 2022) due to the heterogeneous and unpredictable nature of human behavior. However, the entire paradigm of **offline population generation** faces a fundamental limitation: since it relies on model-free reinforcement learning to train cooperative agents, its effectiveness heavily depends on the diversity of the synthesized population, making it challenging to adapt to out-of-distribution partners. DIZCO proposes a fundamental shift in the role of the generative model. Instead of using it offline to generate a static training population, we use it online as a **partner-conditioned world model** for **search**.

**Planning with Generative Models** Generative models, particularly diffusion models, have recently emerged as a powerful paradigm for planning in complex, high-dimensional domains like robotic manipulation and autonomous driving (Chi et al., 2023; Li et al., 2025). A key challenge for these methods is ensuring that the generated plans are not just plausible, but also optimal, especially in out-of-distribution scenarios. Prior work has largely explored two ways to bridge this gap. One way is to guide the sampling process with model-free methods. For instance, Diffuser (Janner et al., 2022) pioneers planning with a diffusion model by training a separate model to predict the cumulative trajectory reward and uses its gradient to guide the trajectory sampling procedure, while Decision Transformer outputs optimal action sequences by conditioning an auto regressive model on the desired return, past states, and actions (Chen et al., 2021). Another, approached outlined by Netanyahu et al. approaches adapting to out-of-distribution tasks by inversely learning a concept from offline demonstrations (Netanyahu et al., 2024). These prior works focus only on single-agent tasks, whereas we focus on multi-agent coordination. Furthermore, our method allows real-time adaption rather than learning from offline demonstrations. We take inspiration from reward-conditioned diffusion policies by training our reward-to-go-conditioned action proposer and combine it with the world model to generate future trajectories.

**Multi-agent World Modeling** Learned world models—generative models that capture an environment's dynamics have shown great promise capabilities for planning and reasoning (Kaiser et al., 2019; Hao et al., 2023; Alonso et al., 2024). While approaches based on reinforcement learning (RL) can find an optimal policy, they suffer from sample inefficiency. In contrast, world models provide an alternative with better sample efficiency which has demonstrated promising outcomes in single-agent tasks. Extending this paradigm to multi-agent scenarios, however, is a significant challenge, as the model must learn to predict not just the environment's physics, but also the behavior of other agents. Recent work has begun to tackle this. For instance, MADiff uses a diffusion-based model for multi-agent trajectory prediction to learn a decentralized policy (Zhu et al., 2025). while COMBO learns a compositional world model by generating video conditioned on joint actions (Zhang et al., 2024). To our knowledge, DIZCO is the only framework that applies the idea of generative world model to zero-shot coordination, realizing the first ever real time adaptation to novel partners utilizing a combination of a partner-conditioned diffusion-based world model and search.

## 3 PRELIMINARIES

**Multi-agent Coordination.** In cooperative multi-agent environments, effective performance depends not only on an agent's ability to act optimally, but also on its capacity to coordinate with diverse partners, especially with novel partners in the environment. The behavior of the partner effectively becomes part of the dynamics of the environment, requiring the ego agent to reason about its partner's intentions.

A standard way to formalize this is through a Markov game (Littman, 1994), defined by $(n, S, \{\mathcal{A}_i\}, \{O_i\}, T, \mathcal{R})$. $n$ denotes the number of agents; $S$ as the set of state space; $\mathcal{A}_i$ is the action

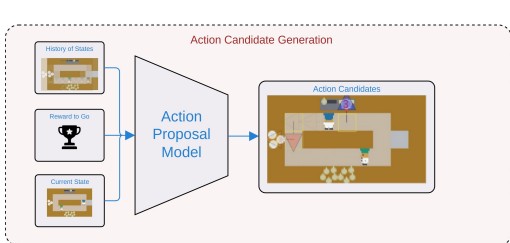

(a) Action Candidate Generation

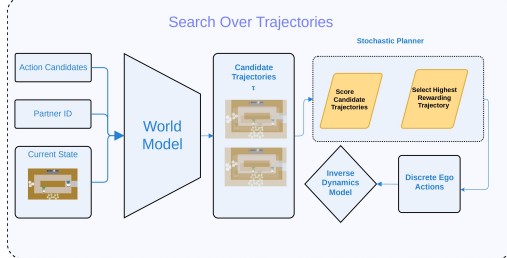

(b) Search Over Trajectories

Figure 2: Overview of the DIZCO framework. **Left**: the action proposal model that generates candidate sequences. **Right**: The world model works with the action proposer to search over trajectories.

space for agent $i$; $O_i$ is the observation space for agent $i$; $T \in S \times A_1 \times ... \times A_n \to \Delta(S)$ is the joint transition model, defining the probability distribution of the next state after taking joint action $a \in A_1 \times ... \times A_n$ in state $s$; $\mathcal{R} \in S \times A_1 \times ... \times A_n \to \mathbb{R}$ is the shared reward function, denoting reward the team receives after taking joint action $a \in A_1 \times ... \times A_n$ in state $s$. In this setting the ego agent controls only its action $a_{\text{ego}}$ while $a_{\text{partner}}$ is determined by an external partner policy $\pi_{\text{partner}}$ which may vary across time and interactions, with the objective of maximizing the expectation of discounted cumulative reward $\sum_{t=0}^{T} \gamma^t r_t$ starting from an initial state $s_0$.

**Planning and Generalizing to N agents** Diffusion models are generative models that can learn distributions over sequences conditioned on context Ho et al. (2020).

Given an initial state $s_0$ and a set of conditioning variables $c = \{c_1, c_2, \ldots, c_k\}$, a diffusion model learns to approximate the distribution over possible future sequences $p_\theta(x_{1:T} \mid x_0, c)$. This is achieved by training a network $\epsilon_\theta$ to predict the injected noise at diffusion step $t$ by minimizing the objective:

$$\mathcal{L}_{\text{MSE}} = \|\epsilon_\theta(x_{1:T}, t \mid x_0, c) - \epsilon\|^2 \tag{1}$$

where $\epsilon \sim \mathcal{N}(0, I)$ and $t$ is sampled uniformly over diffusion steps. At inference time, we employ classifier-free-guidance (CFG) Ho & Salimans (2022) to steer the generation process towards specific condition $c$. The effective noise prediction at each step is the guided combination of a conditional and unconditional prediction:

$$\hat{\epsilon}_\theta(x_t, t, c) = \epsilon_\theta(x_t, t \mid \varnothing) + \omega(\epsilon_\theta(x_t, t \mid c) - \epsilon_\theta(x_t, t \mid \varnothing)), \tag{2}$$

where $\omega$ is the guidance scale, $\epsilon_\theta(x_t, t \mid c)$ is the conditional prediction, and $\epsilon_\theta(x_t, t \mid \varnothing)$ is the unconditional prediction with the conditions dropped. Liu et al. (2022); Netanyahu et al. (2024) shows that we can further combine multiple conditions at once by sampling from the effective composed noise prediction,

$$\hat{\epsilon}_\theta(x_t, t, c) = \epsilon_\theta(x_t, t \mid \varnothing) + \sum_k \omega_k(\epsilon_\theta(x_t, t \mid c_k) - \epsilon_\theta(x_t, t \mid \varnothing)). \tag{3}$$

enabling us to sample a trajectory $x_{1:T}$ that jointly satisfies all conditioning variables $c = \{c_1, c_2, \ldots, c_k\}$, allowing our model to simulate at test time a trajectory $x_{1:T}$ conditioned with a larger number of conditions than those seen at training time, such a larger number of opponents.

## 4 DIZCO: DIFFUSION-BASED ZERO-SHOT COORDINATION

We introduce **DIZCO**, a diffusion based framework for real time zero-shot coordination. The framework is built around two complementary diffusion models operating directly on high-dimensional, image-based observations.

**Offline Dataset** Both diffusion models are trained on an offline dataset of joint trajectories. We generate a dataset by rolling out an expert Cooperator policy (i.e. policy that best cooperates with a partner) $\{\pi_{expert}\}$ trained using reinforcement learning (Liang et al., 2024) with a set of simulated Population-based Training agents as partners. For the simulated agent population $\{\pi_1, ..., \pi_N\}$, we test two conditions: 1) using the same PBT training dataset based on MEP agents as in prior work (Liang et al., 2024) to facilitate comparison, and 2) using a larger dataset with PBT data

from several baseline methods, including Fictitious Co-Play (FCP) (Strouse et al., 2021), maximum entropy population (MEP) (Zhao et al., 2023) and CoMeDi (Sarkar et al., 2023), to test how well our approach scales to larger datasets.

## 4.1 DIZCO

**Partner Conditioned World Model:** The core of **DIZCO** is a diffusion based world model, $\mathcal{G}_\theta$, that learns the joint dynamics of the environment and a partner. Our world model operates directly on image-based observations, learning to predict future frames conditioned on the ego-agent's actions and a partner policy. This disentangles the environment's transition function from any single partner's behavior, a critical advantage over model-free methods that often conflate the two. We model the ego-agent's transition under a fixed partner policy $\pi_{\text{partner}}$ as

$$p(s' \mid s, a_{\text{ego}}; \pi_{\text{partner}}) = \sum_{a_{\text{partner}}} T\big(s' \mid s, a_{\text{ego}}, a_{\text{partner}}\big)\, \pi_{\text{partner}}\big(a_{\text{partner}} \mid s\big),$$

where $T$ is the true environment transition conditioned on the joint actions. Our world model $\mathcal{G}_\theta$ is trained to approximate this distribution across time steps generating videos of future trajectories:

$$\mathcal{G}_\theta\big(s_{t+1:t+H} \mid s_t, a_{\text{ego},t:t+H}, I_{\text{partner}}\big) \approx p\big(s_{t+1:t+H} \mid s_t, a_{\text{ego},t:t+H}, \pi_{\text{partner}}\big),$$

where $s_t$ is the current image, $a_{\text{ego},t:t+H}$ is the $H$-step action plan, $I_{\text{partner}}$ is the partner identity embedding and $s_{t+1:t+H}$ is the predicted video sequence. During training we drop the conditioning elements on the world model, encouraging the model to learn the underlying environment dynamics invariant to any partner policy $\pi_{\text{partner}}$.

**Action Proposal Model:** To generate high-quality action candidate plans for evaluation, we train a separate action diffusion model, $\mathcal{A}_\phi$, to serve as our action proposer. This model is trained on the same offline dataset to generate plausible future ego-action sequences $a_{t:t+K}$. The model is conditioned on a history of $N$ prior states, the current state $s_t$, and reward to go $r_{rtg}$. Conditioning on historical states allows the model to propose actions that are coherent with recent partner behavior, while the reward-to-go conditioning steers proposals towards high-reward outcomes.

**Search Over Trajectories:** Search begins with the action proposer $\mathcal{A}_\phi$ generating $M$ candidate ego-action sequences. For each candidate plan, the world model $\mathcal{G}_\theta$ rolls out $J$ video trajectories. Each simulated trajectory $\tau$ is scored using a deterministic reward function $R(s_t, s_{t+1})$ that operates on pairs of image states. The trajectory $\tau^*$ with the highest cumulative joint partner reward is chosen as the agent's plan.

**Inverse Dynamics Planner:** The final step is to translate the selected visual plan, $\tau^* = (\hat{s_1}, ..., \hat{s_H})$ into discrete environment actions. For this, we utilize an inverse dynamics planner similar to Ajay et al. (2023), that finds the best fit joint action that transitions from an initial state $s_t$ to $s_{t+1}$. The module identifies the action $a^*_{ego}$ that when executed results in closest successor state $\hat{s_{t+1}}$ that matches the true $s_{t+1}$. We convert the full $\tau^*$ trajectories into sequences of ego actions and execute it into our environment. The complete, multi-stage planning process is formalized in Algorithm 1.

**Compositional Generalization to N-Agent Scenarios.** A critical challenge in multi-agent modeling is scalability, as training a unique model for every possible number of agents, $N$ is intractable. Our framework addresses this by leveraging the compositional properties of diffusion models. Our world model $\epsilon_\theta$ is trained to approximate noise of a future trajectory of the controlled vehicle conditioned on on the composite state of all agents in the same environment.

$$\mathbb{E}_{(\tau,c)\sim D_{\text{pretrain}}}[\|\epsilon - (\epsilon_\theta(x_t(\tilde{\tau}), s_0, t|\varnothing) + \sum_{k=1}^{2}(\epsilon_\theta(x_t(\tilde{\tau}), s_0, t) - \epsilon_\theta(x_t(\tilde{\tau}), s_0, t|\varnothing)))\|^2] \quad (4)$$

Given a trained denoising network $\epsilon_\theta$, we adapt this model to environment with N vehicles ($N > 2$) by composing the conditions of $N$ vehicles.

$$\hat{\epsilon}(\epsilon_\theta) = \epsilon_\theta(x_t, s_0, t|\varnothing) + \sum_{N=1}^{N}(\epsilon_\theta(x_t, s_0, t) - \epsilon_\theta(x_t, s_0, t|\varnothing))) \quad (5)$$

We evaluate the behavior it generates by initializing $x_T(\tau) \sim \mathcal{N}(0, \alpha\mathbf{I})$, and compute $x_t$ iteratively as a function of the estimated denoising function $\hat{\epsilon}(\epsilon_\theta)$ until generating $x_0 = \tau$ representing the trajectory of the controlled vehicle. We use an inverse planner to produce an action given a current state and a future state in the predicted trajectory.

## 4.2 REAL-TIME PLANNING ARCHITECTURE

A fundamental barrier to using large, generative video models for real-time interaction is simply their significant computational cost. In a dynamic, real-time interactive environment, a naive "stop-and-plan" approach is simply non-viable. While the agent pauses to compute its next move, the partner continues to act and the world continues to change, rendering the resulting plan obsolete before it can even be executed. To solve this critical problem, we designed DIZCO to be completely capable of executing asynchronously, detaching planning from execution.

As shown in Figure 8, an **Agent Interface** manages the the system's core component: an **Action Queue**, enabling the following two parallel processes:

- **The Execution Loop (Low-Latency):** On every environment tick, the interface immediately dequeues a pre-computed action from the buffer. This ensures the agent is always acting on the most recent plan and remains responsive to the live game state.
- **The Planning Loop (Asynchronous):** In a separate background process, the full DIZCO planning engine runs continuously. It takes the latest available state, proposes candidate plans, and uses its world model to search for an optimal action sequence. This newly generated plan then entirely replaces the previous action buffer, ensuring the agent's strategy is continually refreshed based on up-to-date information.

**Validating Real-Time Performance.** This architecture is a practical and elegant solution that proved highly effective on our testbed (a single NVIDIA RTX 4090 GPU shared between models). Despite the high temporal resolution of our evaluation domain, our asynchronous design successfully hides the planner's latency, allowing us to deploy our world model (133.46M parameters) and action proposal model (168.89M parameters) at their full capability, while ensuring the agent's actions were always based on a recently computed and relevant plan. Our architecture demonstrates a robust solution for deploying computationally intensive generative planners in dynamic, real-time interactive settings.

## 5 EXPERIMENTS

Our experiments are designed to validate the two core capabilities of diffusion models outlined in our introduction: **1) adaptation to novel partners** in complex, human-AI coordination tasks, and **2) generalization to an arbitrary number of agents**. We first detail our experimental setup before presenting the results that substantiate these two central claims.

**Generalization of the World Model to $N$ Agents:** To isolate and evaluate the compositional generalization of our world model, we use the Autonomous Driving domain (Leurent, 2018a), an agent acts in a challenging multi-agent environment to complete a driving task. For this experiment, We test this capability by training our world model exclusively on two-agent interactions and then evaluating its zero-shot planning performance in more crowded environments with $N > 2$ agents. The world model, $\epsilon_\theta$, is trained on a dataset of interactions where each scene contains a fixed number, $K$, of other agents. The offline expert dataset was collected using a deterministic tree search planner provided in Leurent (2018b) . Each vehicle is represented by a 7-tuple, including whether it is presented on the road, its x and y positions and velocities, and cosine and sine heading directions. The agent's task is a high-speed highway merge, with performance assessed via quantitative metrics (mean reward, total distance, and collision rate). This setup provides a benchmark to quantify a world model's ability to scale its reasoning to more complex multi-agent interactions.

**Adaptation To Novel Partners:** We evaluate DIZCO in the **Overcooked** domain, a standard benchmark for human-AI cooperation. The environment requires two players to coordinate to complete cooking tasks, demanding the anticipation of partner goals and adaptive behavior to succeed. We focus on two challenging layouts: **Counter Circuit**, which tests coordination in navigation and object passing, and **Multi-Strategy** (Diverse Counter Circuit) which features a combinatorially expanded strategy space that makes accurate partner inference essential to success. Our main offline

| Method | Counter Circuit | Multi-strategy |
|--------|-----------------|----------------|
| **DIZCO** | $\mathbf{122.0 \pm 40.93}$ | $\mathbf{104.0 \pm 51.22}$ |
| Action Proposal Model Only | $56.8 \pm 30.03$ | $39.2 \pm 35.32$ |
| GAMMA HA FFT | $91.84 \pm 26.03$ | $34.05 \pm 17.98$ |
| CoMeDi | $69.62 \pm 29.61$ | $32.02 \pm 16.90$ |
| MEP | $76.19 \pm 16.90$ | $64.02 \pm 18.51$ |
| FCP | $32.11 \pm 13.42$ | $44.22 \pm 19.23$ |

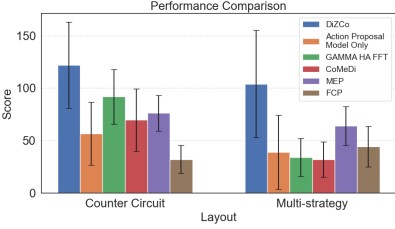

Figure 3: **DIZCO's State-of-the-Art Performance and Component Analysis.** Main performance table comparing DIZCO variants against prior SOTA methods on the held-out human proxy.

evaluation is conducted against a held out Human Proxy model, a behavior-cloned (BC) policy from human game play data as in prior work (Carroll et al., 2019). The Human Proxy model allows for automatic and cheap evaluation of agents' performance. Our results demonstrate that the performance of the full **DIZCO** significantly outperforms prior state-of-the-art baselines in zero shot coordination.

## 5.1 EVALUATION CRITERIA

We further analyze the performance of **DIZCO** according to the following criteria:

- **Generalization to Multiple Agents.** How well does the world model adapt to environments with different numbers of agents in the **Autonomous Driving Domain** ?
- **End-to-End Agent Performance.** How well does the full **DIZCO** perform against prior baselines when paired with the human proxy model ?
- **World Model Accuracy & Controllability.** How faithfully does the world model simulate partner-conditioned video trajectories compared to ground truth roll outs ?
- **Action Proposal.** How effective is the action proposer at generating high-quality, cooperative plans as measured by the performance against the human proxy model?

## 6 RESULTS

**Compositional Generalization to N-Agent Scenarios:** A key advantage of our generative approach is the potential for compositional generalization. To test this, we evaluate whether our world model, trained exclusively on two-agent scenarios in the Autonomous Driving Domain, can successfully simulate and plan in more crowded environments with $N > 2$ agents. As shown in Figure 6, the world model is demonstrates impressive and consistent generalization. Despite having never seen multi-agent scenarios during training, it maintains high performance across all three evaluation metrics (mean reward, total distance traveled, and average speed) even as the number of vehicles increases to eight. This result serves as a strong proof-of-concept for the compositional reasoning capabilities of our generative world model, a promising feature for scaling up to more complex, real-world coordination tasks.

**Coordinating with a Novel, Held-Out Partner:** Having validated the generalization capabilities of our world model, we now turn to our primary evaluation: DIZCO's ability to coordinate with a novel, held-out partner in a true zero-shot setting. We test this against the human proxy model trained on held-out test data, $\pi_{proxy}$ as our partner like prior work. The results (Table **??**) clearly demonstrate that **DIZCO establish a new state-of-the-art**, outperforming 4 competitive baselines, including GAMMA, the prior state-of-the-art. Notably, DIZCO's performance is most pronounced on the complex **Multi-strategy** layout. This layout features a significantly larger and more ambiguous strategy space, which seems to be a major failure point for the purely reactive, model-free baselines. In contrast, we hypothesize DIZCO was better able to navigate this complexity, identify higher-rewarding joint plans, and thereby coordinate more effectively than baselines.

## 6.1 LIVE HUMAN EVALUATION

We conducted a preliminary user study designed to test the validity of the asynchronous architecture in a live interactive setting and further understand how humans perceive the agent's model-based behavior.

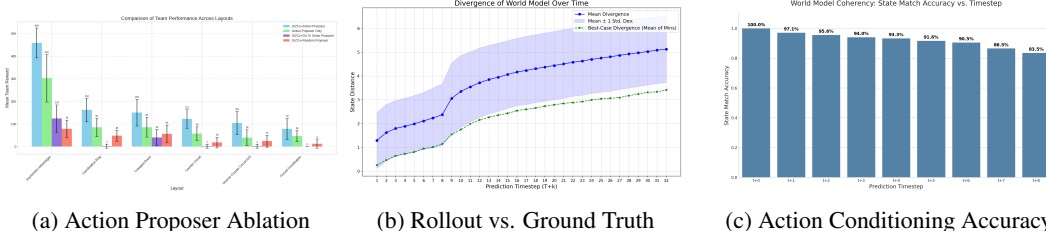

(a) Action Proposer Ablation    (b) Rollout vs. Ground Truth    (c) Action Conditioning Accuracy

Figure 4: **Action Proposer and World Model Ablation Study.** (a) **Action Proposer Effectiveness**: Ablation study showing the contribution of a learned action proposer compared to simpler heuristic alternatives. (b) **Long-Horizon Consistency:** Predictive error (state mismatches vs. ground truth) over a 32-step rollout. The model maintains high accuracy over short horizons, generating plausible, non-collapsing futures across the full trajectory. (c) **Action Controllability:** The model's accuracy in producing a successor states that correctly corresponds to a given conditioning actions, confirming its reliability as a simulator for planning.

**User Feedback:** The feedback indicated that DıZCo was a fairly effective and cooperative partner. From all the feedback received in C, we identified two primary themes:

- **Reliability:** Participants described the agent as being a dependable contributor, using phrases like "a steady partner" and remarking it "contributed reliably without needing constant correction."
- **Adaptiveness:** More importantly, users viewed DıZCo as being adaptive to their intents, with one user remarking "It wasn't perfect, but it showed signs of **adapting** as the game progressed" and another: "The agent was **cooperative**, and that made everything entertaining".

## 6.2 ABLATIONS

To understand the key drivers of DıZCo's performance, we conducted a series of targeted evaluations on its core components.

**World Model Accuracy & Controllability:** The entire DıZCo framework is contingent on the quality of its learned world model. If the simulator is inaccurate, the search process will optimize for the wrong futures. To validate this, we evaluate the world model on two criteria: its long-horizon consistency with reality and its controllability via action conditioning.

As shown in Figure 4, our world model demonstrates high fidelity. The roll-out consistency (Figure 4b) is nearly perfect over short horizons (1-8 steps). While predictive errors naturally accumulate, the model still generates plausible, non-collapsing trajectories over the full 32-step horizon. Furthermore, the model proves to be highly controllable (Figure 4c), reliably producing future states corresponding to its given conditioning action. These evaluations confirm our world model serves as an effective and trustworthy simulator, providing a solid foundation for search-based planning.

**Search Budget:** As shown in Figure 5a, performance scales directly with the search budget, the number of candidate plans and simulations per plan. A minimal budget results in poor performance, but we see significant immediate gains as we increase the number of candidates, allowing for more diverse plans to be considered. Increasing the number of simulations per candidate, further increases performance but it is best paired with more action candidates. Performance gains eventually plateau, indicating there is a point of diminishing returns with search.

**Planning Horizon:** Figure 5b reveals a crucial insight: shorter planning horizons (e.g. 8-16 steps) consistently outperform longer ones for all evaluation partners. This result does not imply that planning is fundamentally unsuited for highly dynamic, real-time domains; rather it highlights a common failure mode of long open-loop plans when partner actions are stochastic.

## 6.3 THE IMPORTANCE OF A LEARNED ACTION PROPOSER

The success of the DıZCo framework relies on the synergy between its search procedure and action proposer. We conduct two experiments to validate the contribution of the diffusion-based proposer.

**Performance in Isolation:** We first establish a baseline performance by evaluating the Action Proposal Model in isolation from world model simulation or search. As shown in Table **??**, the performance of the proposer alone is modest. While it is able to generate plausible strategies, it is

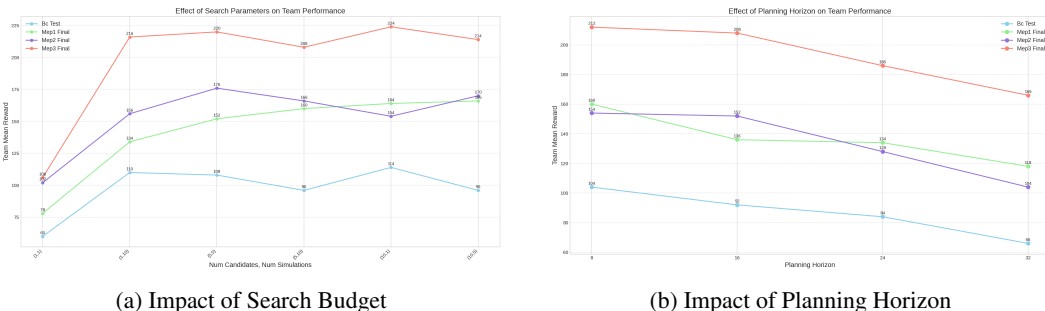

(a) Impact of Search Budget        (b) Impact of Planning Horizon

Figure 5: **Hyperparameter Sensitivity Analysis.** (a) **Search Budget vs. Performance:** Team reward scales with the number of candidate plans and simulations but plateaus, revealing a point of diminishing returns. (b) **Planning Horizon vs. Performance:** Shorter planning horizons consistently outperform longer ones, highlighting frequent re-planning is a more robust strategy than long-term commitment.

not competitive with top-performing baselines like GAMMA HA FFT on its own, indicating that the proposer's role is not to be a perfect standalone policy, but to provide a diverse set of candidates for the search process to evaluate. The immense performance leap with the world model and search showcases that the explicit process of simulating partner conditioned futures, and performing search over them is the true engine of DIZCO, transforming the modest proposer into a state-of-the-art coordination agent.

**Other Proposer Methods:** Second, to test the quality of our action candidates, we evaluate the full DIZCO framework with two simpler alternative proposers: a deterministic "go to onion" heuristic and a random action policy. As shown in Figure 4a, this substitution leads to dramatic collapse in performance, confirming that the search procedure requires diverse and strategically plausible set of candidate plans to explore.

# 7 LIMITATIONS AND FUTURE WORK

**Reliance On True Reward Functions and Inverse Planners:** The current implementation of **DIZCO** relies on two non-learned components: a deterministic reward function to score simulated trajectories and a brute-force inverse dynamics function to translate visual plans into actions. This design choice was made to improve reliability with search, however further work will be needed to extend the results to environments where ground-truth functions are not available.

**Challenges of the Asynchronous Architecture:** Leveraging large, computationally intensive diffusion models for planning necessitates our asynchronous design. However, this introduces its own fundamental trade-off. While the architecture successfully hides planning latency, it creates a temporal gap between when a plan is computed and when it is fully executed. This can lead to moments of coordination friction, where the agent is committed to a pre-computed plan that has become stale due to a partner's split-second action, as observed by one of our users in the study: 'DIZCO was able to recognize when I was going to grab the soup or pick up onions I had placed down, but there were still moments of conflict or inefficiency." Improving the core efficiency of the generative models to shrink this temporal gap is a critical challenge for future of real-time, model-based agents.

# 8 CONCLUSION

We introduced DIZCO, a framework that recasts zero-shot coordination as a problem of online, search-based planning with generative models. Our method achieves state-of-the-art performance by explicitly inferring a partner's policy and simulating future outcomes to select an optimal plan, revealing that a model-based approach, which can reason about partner uncertainty, is fundamentally more robust than prior model-free methods. We believe this paradigm of generative planning and inference is a critical step towards creating truly adaptive agents capable of human-AI coordination.

## 9 ETHICS STATEMENT

This research aims to develop the next generation of AI agents capable of zero-shot coordinate with novel partners in real-time, addressing an important challenge in the field of Cooperative AI (Dafoe et al., 2020). To evaluate our approach, we conduct human evaluation of our method with human participants recruited from the Prolific, an online crowdsourcing platform following an IRB-approved protocol. Our work seeks to develop the first open-sourced framework that leverages generative models to enable real-time, search-based planning in a complex human-AI cooperative task. By releasing this framework, we aim to facilitate its adaptation to other cooperative domains, helping to further improve the collaborative capabilities of AI agents.

## 10 REPRODUCIBILITY

To promote reproducibility, we provide our source code and corresponding configuration files used for running the experiments in open source. The details about installation and sample code for running the experiment will be included at the project page.

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

# 11 APPENDIX

## A AUTONOMOUS DRIVING DOMAIN RESULTS

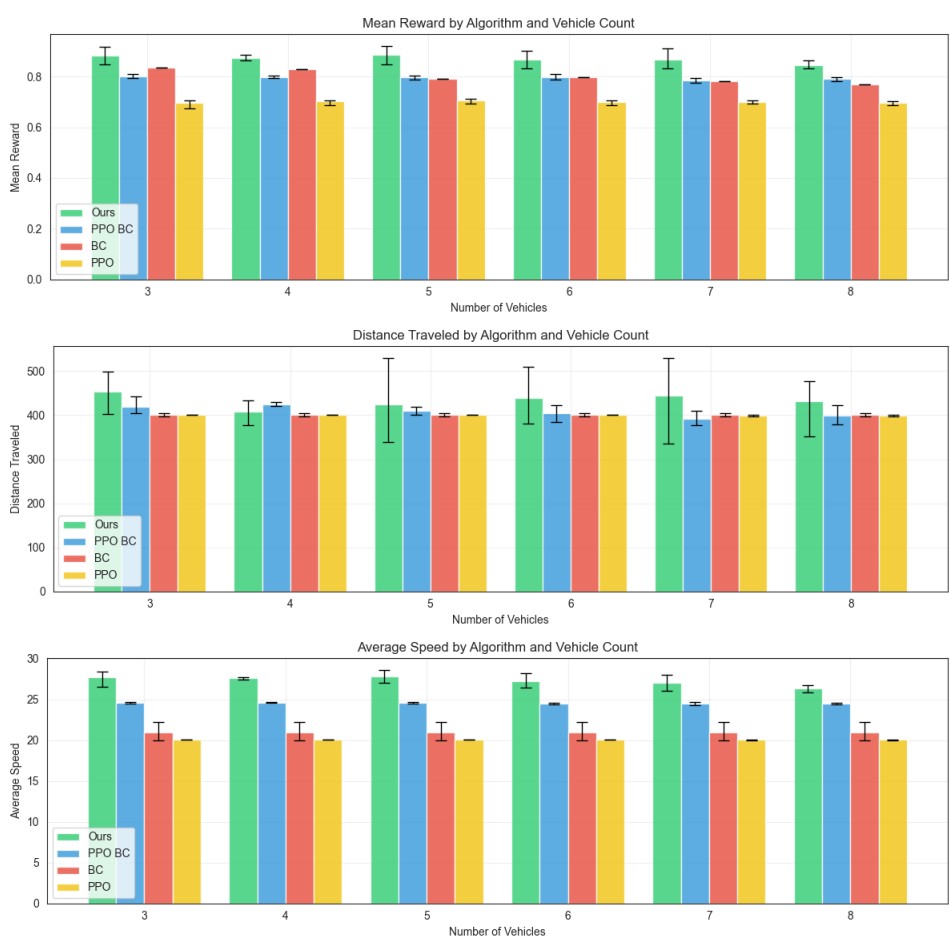

Figure 6: **Autonomous Driving domain comparison.** Performance of diffusion-based world model generalization comparing to baseline algorithms. We report standard errors over 5 trajectories. Overall, our method learns to generalize to environments with increasing number of agents with competitive mean reward, total distance traveled and average speed.

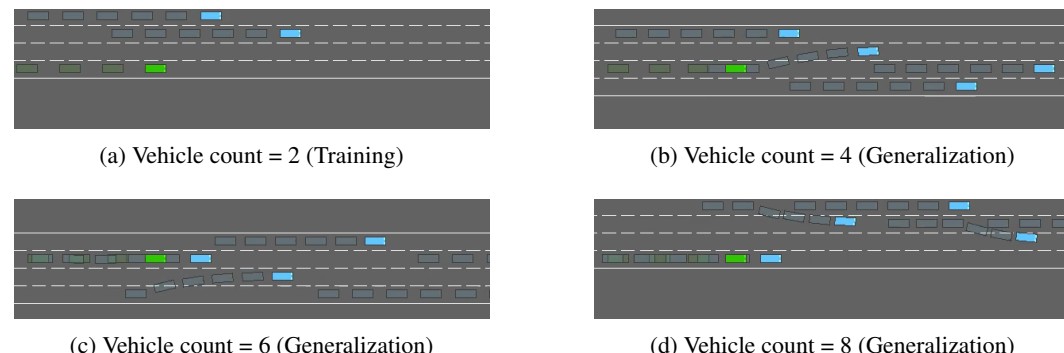

(a) Vehicle count = 2 (Training)  (b) Vehicle count = 4 (Generalization)

(c) Vehicle count = 6 (Generalization)  (d) Vehicle count = 8 (Generalization)

Figure 7: Screenshots of training and generalization of autonomous driving domain. Our world model is trained on the environment with vehicle count as 2, and we generalize to environments with 3-8 vehicles. Some vehicles are not visible in the screenshot due to screen width limitation.

## B  REAL TIME EVALUATION ARCHITECTURE

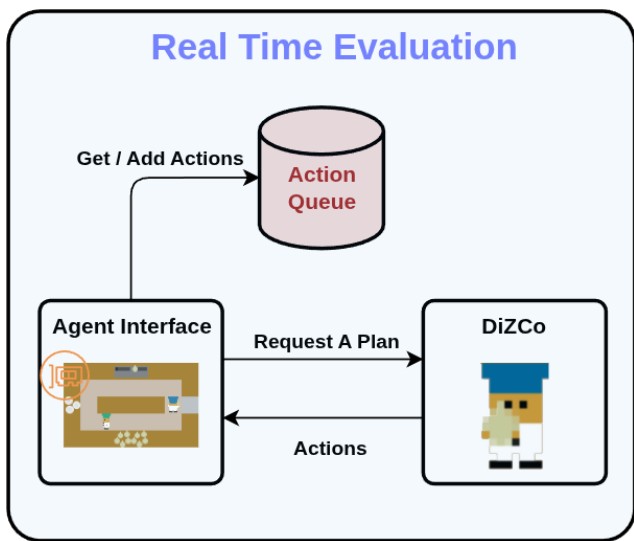

Figure 8: The **DIZCO** Architecture for Real-Time Human Interaction. To overcome the high latency of the generative planner, we use an asynchronous design managed by an **Agent Interface**. (1) A buffered **Action Queue** stores a sequence of pre-computed actions. (2) On every environment step, the interface immediately dequeues the next action for a low-latency response. (3) In the background, when the queue falls below a threshold, the full **DIZCO Planner** is triggered to generate a fresh, updated plan, which then replaces the buffer.

## C  QUALITATIVE RESULTS OF HUMAN EVALUATION

As mentioned in 6.1, individuals consistently reported that DIZCO demonstrated sign of adapatibility and responsiveness. We provide additional participant feedback for DIZCO agents from the user study as follows:

- "I found this agent to be a steady partner. It made an effort to sync with my movements and stuck to a predictable rhythm, which helped me plan ahead. While it occasionally lingered near key areas, it didn't cause major disruptions. Its behavior felt somewhat human-like, and I enjoyed the collaboration overall. It contributed reliably without needing constant correction."

- "I noticed this agent made a few attempts to sync with my movements, especially when I was at the pot or serving counter. It wasn't perfect, but it showed signs of adapting as the game progressed. Genuinely, my experience with this agent was solid — it played its role reliably, let me take the lead on strategy, and didn't overcomplicated things, which worked well for a fast-paced kitchen."

- "The agent was responsive and made the game interaction smooth and engaging."

- "The agent was cooperative and that made everything entertaining."

- "I had a good interaction with the agent and that was awesome."

- "Able to grab onions I put down, recognize when I was going to grab the soup, etc. However, still felt it did not exactly coordinate with me, still end up in situations where we are in conflict or both stuck."

## D  DIZCO ALGORITHM

---

**Algorithm 1** The **DIZCO** Online Planning Algorithm

---

1: **Input:** State $s_t$, history $h_t$, world model $\mathcal{G}_\theta$, action proposer $\mathcal{A}_\phi$, number of candidate $I$, number of simulations per candidate $J$, fixed future state horizon $H_{fixed}$, planning horizon $H_{plan} \leq H_{fixed}$, reward function $R(s, s')$, inverse-dynamics function $D(s, s')$, partner ID $I_{partner}$.

2: **Propose a set of candidate action plans of horizon** $H_{fixed}$
3: $\{\mathbf{a}^{(1)}, ..., \mathbf{a}^{(I)}\} \leftarrow \mathcal{A}_\phi(s_t, h_t, r_{rtg}, H)$

4: **Search: Simulate and Evaluate Trajectories**
5: Initialize $\mathcal{T} \leftarrow \emptyset$
6: **for** $i = 1$ to $I$ **do**
7:     **for** $j = 1$ to $J$ **do**
8:         $\tau^{(i,j)} \leftarrow \mathcal{G}_\theta(s_t, \mathbf{a}^{(i)}, I_{\text{partner}})$
9:         $R^{(i,j)} \leftarrow \sum_{k=0}^{H-1} F(\tau_k^{(i,j)}, \tau_{k+1}^{(i,j)})$     where $\tau_0^{(i,j)} = s_t$
10:         $\mathcal{T} \leftarrow \mathcal{T} \cup \{(\tau^{(i,j)}, R^{(i,j)})\}$
11:     **end for**
12: **end for**

13: **Select Plan and Ground Action**
14: $\tau^* \leftarrow \arg\max_{(\tau,R)\in\mathcal{T}} R$
15: $a_t^* \leftarrow D(s_t, \tau^*)$

16: **Execute only first** $H_{plan}$ **actions of the best plan**
17: **return** $\mathbf{a}_{1:H}^*$

---

## E  HYPER-PARAMETERS

Below we outline our selected hyper-parameters for DIZCO's use cases.

### E.1  ARCHITECTURE

Table 1: Architecture Parameters.

| Hyperparameter | World Model (Overcooked) | Action Proposer | World Model (Driving) |
|---|---|---|---|
| Num. Parameters | 133.46M | 168.89M | 11.69M |
| Input Resolution | $9 \times 5$ | $9 \times 5$ | $7 \times 1$ |
| Output Resolution | $9 \times 5$ | $1 \times 32$ | $7 \times 1$ |
| Base Channels | 256 | 128 | 128 |
| Num. Res. Blocks | 3 | 2 | 2 |
| Attention Resolutions | $(1, 2)$ | $(4, 8)$ | $(4, 8)$ |
| Channel Multipliers | $(1, 2)$ | $(1, 2, 4, 8)$ | $(1, 2, 4, 8)$ |

## E.2 TRAINING

Table 2: Training evaluation parameters for DIZCO

| Hyperparameter | World Model (Overcooked) | Action Proposer | World Model (Driving) |
|---|---|---|---|
| Batch Size | 64 | 64 | 64 |
| Learning Rate | $1e^{-4}$ | $1e^{-4}$ | $1e^{-4}$ |
| CFG Dropout | 0.1 | 0.1 | 0.1 |
| Training Steps | 600,000 | 600,000 | 800,000 |
| Num Partner Policies | 27 | 27 | NaN |
| Compute | NVIDIA H200 | NVIDIA H200 | NVIDIA L40 |

## E.3 ONLINE PLANNING

Table 3: Hyper-parameters for **DIZCO** Online Planning.

| Parameter | Symbol | Value | Description |
|---|---|---|---|
| Number of candidates | $I$ | 10 | Number of action plans proposed by $\mathcal{A}_\phi$. |
| Simulations per candidate | $J$ | 5 | Number of forward simulations per candidate plan. |
| Fixed future horizon | $H_{\text{fixed}}$ | 32 | Maximum horizon length for action proposals. |
| Planning horizon | $H_{\text{plan}}$ | 8 | Number of executed steps ($\leq H_{\text{fixed}}$). |
| Reward-to-Go | $r_{\text{rtg}}$ | 150.0 | Reward-To-Go Target for Action Proposer |

## E.4 REAL-TIME EVALUATION

Table 4: Hyperparameters for **DIZCO** real-time evaluation.

| Parameter | Symbol | Value | Description |
|---|---|---|---|
| Number of candidates | $I$ | 5 | Number of action plans proposed by $\mathcal{A}_\phi$. |
| Simulations per candidate | $J$ | 2 | Number of forward simulations per candidate plan. |
| Action queue size | $H_{\text{fixed}}$ | 32 | Maximum length of the action queue. |
| Planning horizon | $H_{\text{plan}}$ | 8 | Number of executed steps ($\leq H_{\text{fixed}}$). |
| Reward-to-go target | $r_{\text{rtg}}$ | 150.0 | Target reward-to-go for action proposer. |

## F USE OF LARGE LANGUAGE MODELS (LLMS)

LLMs were used solely for language polishing and improving the clarity of writing in this paper. They were not involved in the research design, ideation, analysis, or generation of scientific content.

