# OpenReview forum: "DiZCo: Planning Zero-Shot Coordination in World Models"
_ICLR.cc/2026/Conference — ICLR 2026 Conference Withdrawn Submission_

### Official Review · Reviewer_nVJz · 2025-10-31

**Soundness:** 2
**Presentation:** 2
**Contribution:** 1
**Rating:** 4
**Confidence:** 4

**Summary:**

This work introduces DIZCO, a framework that utilizes generative models to enable real-time, search-based planning in complex human-AI cooperative tasks. DIZCO is a model-based method that employs a generative model (the world model) to predict future world trajectories conditioned on the current state, the ego agent's actions, and the partner's identity. Offline evaluations indicate that the DIZCO framework outperforms state-of-the-art model-free policies in terms of joint reward.

**Strengths:**

The work operates in an offline setting, leveraging rollouts sampled from a previous population-based method. This is a significant strength as it eliminates the need for human labeling and substantially reduces the reliance on costly RL sampling.

The experimental results clearly demonstrate superior performance when compared to previous population- based training methods.

**Weaknesses:**

1. The claim in line 20, “the first framework that leverages generative models to enable real-time…,” appears to be an overclaim. Existing works, such as Proagent [1] and E3T [2], also focus on using generative models or related concepts (like partner prediction modules) to address human-AI cooperative tasks. The authors should specify the unique differentiating characteristic of DIZCO's use of generative models compared to prior art.
[1]Zhang, Ceyao, et al. "Proagent: building proactive cooperative agents with large language models." Proceedings of the AAAI Conference on Artificial Intelligence. Vol. 38. No. 16. 2024.

[2] Yan, Xue, et al. "An efficient end-to-end training approach for zero-shot human-AI coordination." Advances in neural information processing systems 36 (2023): 2636-2658.

2. The motivation for using a diffusion model for planning and generalizing to $N$ agents is unclear. It appears to be primarily used for sequential state prediction.

3. The paper mainly describes the methodology for the two-player setting. To justify the highlight on the $N>2$ agents setting, the authors may need to re-examine and potentially clarify the formulation (e.g., in line 229) to better accommodate and demonstrate generalizability to $N$ agents.

4. It would be better to demonstrate the necessity of the diffusion model by comparing its performance against other powerful neural network architectures (such as CNNs or RNNs/LSTMs) or computing the world model prediction precision.

5. The "Partner Conditioned World Model" relies only on the predicted ego action sequences and the partner's identity. There is a concern regarding prediction precision due to the cumulative error from both sequential action prediction (via the Action Proposal Model) and trajectory prediction.

In general, the overall workflow of the method is too complex and could benefit from a clearer and straightforward motivation for each component.

**Questions:**

Typos: There appears to be a missing closing parenthesis ')' in line 185.

In line 229, the text mentions "a fixed partner policy $\pi_{\text{partner}}$." Please clarify which specific action policy plays the role of $\pi_{\text{partner}}$ in the overall framework?

The paper mentions computing a deterministic reward function $R(s_t, s_{t+1})$. Given that the states ($s_t, s_{t+1}$) are generated by a diffusion model (which is stochastic), please clarify if the reward function $R$ is itself a separate trainable model?

The workflow involving the Inverse Dynamics Planner is confusing. The model first generates full $\tau^*$ trajectories based on action proposals, and then the Inverse Dynamics Planner is used to "convert the full $\tau^*$ trajectories into sequences of ego actions and execute it into our environment." If the trajectories are already generated conditioned on action proposals, why is a separate Inverse Dynamics Planner necessary to re-convert the trajectory back into actions? This step makes the overall workflow seem overly complex and lacks clear motivation. A clearer explanation of its role is needed.

The word size in the figures is too small, making them difficult to read. This should be corrected for readability.

---

### Official Review · Reviewer_3vb8 · 2025-10-31

**Soundness:** 2
**Presentation:** 2
**Contribution:** 2
**Rating:** 2
**Confidence:** 5

**Summary:**

DIZCO combines a partner‑conditioned diffusion world model with a diffusion‑based action proposer. At test time it generates candidate actions, simulates their futures in the learned world model, and searches over the rollouts to select a plan, enabling adaptation to novel partners and outperforming SOTA baselines in Overcooked. It further demonstrates 2→N compositional generalization via composable CFG, and presents an asynchronous architecture for online interaction. The method and system are mutually reinforcing, and the experiments cover end‑to‑end performance plus key ablations.

**Strengths:**

1.Clear factorization. The approach cleanly decouples partner/environment modeling from planning/search, aligning with the test‑time compute paradigm for reasoning.

2.End‑to‑end gains. On challenging layouts with a Human Proxy, DIZCO significantly outperforms baselines; the “action‑proposer only” variant lags, underscoring the central role of world‑model + search.

3.Compositional generalization. Using CFG to add conditions enables zero‑shot planning for N>2 agents—preliminary but directionally novel.

**Weaknesses:**

1.Benchmark scope. Overcooked is relatively constrained for human‑AI collaboration. Stronger evidence on richer variants (e.g., Overcooked‑V2) or additional coordination suites is needed.

2.Result prioritization. The placement is suboptimal: Sec. 6.1 is underpowered and fails to make the method’s effectiveness unmistakable, while several important results are pushed to the appendix.

3.Exposition depth. Given the breadth (world models, diffusion policies, compositional guidance), both the main text and appendix are too terse, making it hard to reproduce or scrutinize design choices.

4.Live‑collaboration claim under‑supported. The core claim is real‑time human–AI collaboration, yet quantitative evaluation relies on a Human Proxy; the live user study is qualitative only (no sample size, controls, or statistics), so the claim risks overstatement.

**Questions:**

1.Human Proxy fidelity. What are the data composition, scale, and style diversity for the proxy? How closely does it match real human behavior, and what gaps are measured?

2.Embodied collaboration. Can the method transfer to embodied robotics settings? If so, what changes are required for reward specification, action mapping, latency control, and safety[1]?

3.Diversity of teammates. How do you quantify the diversity of generated partner behaviors or joint plans? Without sufficient diversity, how is compositional generalization justified[2]?

4.Why it works (toy example). Provide a minimal, visual toy example that shows how test‑time search over world‑model rollouts corrects a suboptimal initial proposal.

5.Limits of composable CFG. As N increases or interactions strengthen, does additive composition introduce conflicts or inconsistencies? Show failure cases or a breakdown curve vs. N.

6.Why diffusion for the world model? What concrete advantages over autoregressive/latent‑trajectory alternatives under compute‑matched search budgets?

Ref:

[1] Multi-agent embodied ai: Advances and future directions

[2] Learning to Coordinate with Anyone

---

### Official Review · Reviewer_rUkP · 2025-11-01

**Soundness:** 2
**Presentation:** 2
**Contribution:** 3
**Rating:** 4
**Confidence:** 4

**Summary:**

This paper introduces DiZCo, a diffusion-based framework for zero-shot coordination in human-AI cooperative tasks. The core idea is to leverage generative models for real-time, search-based planning. The authors propose a two-model architecture: an action proposer that generates candidate action sequences and a partner-conditioned world model that simulates future trajectories. By performing test-time search over these simulated rollouts, the agent can adapt to novel partners without relying on pre-trained policies.

**Strengths:**

The idea of using diffusion-based world models for zero-shot coordination is novel. The compositional generalization of the world model to an arbitrary number of agents is a creative and technically sound contribution. And the real-time human interaction framework could be useful for the community.

**Weaknesses:**

- **Technical Errors and Inconsistencies**:
    - Several typos and formatting issues exist (e.g., line 64: missing Figure 1 reference; line 260: repeated "on"; line 283: repeated "the"; line 358: grammatical error).
    - Equations (4) and (5) are incorrect with conditions missing and poorly explained, especially the sudden introduction of "vehicle" without context.
    - The use of an inverse dynamics planner is not well-justified. If the action diffusion model outputs action sequences, why is an additional inverse planner needed?
    - Missing references for tables and figures (e.g., line 431: missing table reference; line 367: missing figure reference).
- **Experimental Evaluation**: The experiments are insufficiently comprehensive:
    - Autonomous Driving only tests world model generalization, not full decision-making performance.
    - Overcooked is only tested on two layouts, limiting the scope of evaluation.
    - Only one human proxy model is used for evaluation, while previous works [1,2] use multiple models, raising concerns about generalizability.
    - No code is provided, which hinders reproducibility and verification of results.
- **Presentation**:
    - Figures 4 and 5 are too small and illegible, making it difficult to interpret the ablation and sensitivity results.

References

[1] Chao Yu, et al. Learning Zero-Shot Cooperation with Humans, Assuming Humans Are Biased. ICLR 2023.

[2] Lihe Li, et al. LLM-Assisted Semantically Diverse Teammate Generation for Efficient Multi-agent Coordination. ICML 2025.

**Questions:**

1. How is partner identity modeled and integrated into the world model? Is it learned or hand-engineered?
2. How is diversity in action candidates ensured? Is there any explicit diversity-promoting mechanism, or does it rely solely on the stochasticity of the diffusion model?
3. Why is an inverse dynamics planner necessary? If the action proposer outputs action sequences, why not use them directly?

---

### Official Review · Reviewer_5zGh · 2025-11-07

**Soundness:** 2
**Presentation:** 2
**Contribution:** 2
**Rating:** 4
**Confidence:** 3

**Summary:**

This paper tackles the problem of human-AI coordination (e.g. overcooked) and adapting to unseen partners or more agents than were seen during training. The method consists of two diffusion models, one a world model and the other an action proposer, which together define partner-conditioned action plans to maximise joint reward. An asynchronous plan creation vs plan execution mechanism allows the computationally intensive plan creation to span multiple steps while actions are executed according to the latest available plan for low latency.

The paper presents an interesting use of generative models for human-AI coordination in decision-making tasks, as well as a nice asynchronous execution mechanism. Unfortunately, the framing and testing of the algorithm is fundamentally unfair, along with some other issues. Therefore, the paper does not convince me of the methodology and is not competitive as is.

**Strengths:**

* The paper is reasonably clear and the diagrams and graphs look good
* The asynchronous plan creation/execution mechanism is interesting and could be applied to other RL methods, somewhat in line with how humans may take many steps to formulate a plan, while executing according to the latest known plan.
* Compared to the considered baselines, the results are indeed favourable.
* The authors present a decently large set of experiments, including two environments and several ablations.

**Weaknesses:**

* The main issue is that the method unfairly uses privileged information, i.e. expert trajectories to train both the world model and action proposer. By contrast, all considered baselines use RL for the policy. GAMMA indeed uses expert trajectories, but only to model partner identities, which can be considered part of the environment. Its policy is trained without expert demonstrations. So it seems Dizco could just be reproducing high-performing expert trajectories rather than truly generalising. To be convinced of efficacy, I would need to see it outperform state-of-the-art imitation learning baselines and the framing of the method would need to be as an imitation learning algorithm.
* The method is tested on simple 2D environments and already needs the asynchronous mechanism to account for slow test-time computation. I’m concerned that, were it to be extended to more complex domains (e.g. COMBO tested on ThreeDWorld), the computation would be prohibitively slow and therefore the action plan would be very stale.
* Some typos, GAMMA not cited, some table references missing.

**Questions:**

* I didn’t fully understand why the inverse dynamics model was needed? Don’t you already have access to the action sequence that generated the candidate state sequence (since the latter was generated based on the former by the world model)?
* Isn’t this setting an example of a Dec-POMDP rather than a Markov game since the reward is shared?

---

### Note · Authors · 2025-11-25

**Comment:**

We found one key metric reported in our paper is misaligned with the metric reported by the baseline we are comparing to. Therefore we decide to withdraw the paper.

**Withdrawal Confirmation:**

I have read and agree with the venue's withdrawal policy on behalf of myself and my co-authors.